# New Frontiers in Diagnosis and Prevention of Acute Kidney Injury (AKI): The Role of Dendritic Cells and Innovative High-Throughput Techniques

Grazia Maria Virzì [1,2,*] , Anna Clementi [3], Maria Mattiotti [1,2] , Giovanni Giorgio Battaglia [3], Claudio Ronco [2] and Monica Zanella [1,2]

1   Department of Nephrology, Dialysis and Transplant, San Bortolo Hospital, International Renal Research Institute Vicenza, 36100 Vicenza, Italy
2   IRRIV-International Renal Research Institute Vicenza, 36100 Vicenza, Italy
3   U.O.C. Nefrologia e Dialisi, Ospedale Santa Marta e Santa Venera, 95024 Acireale, Italy
*   Correspondence: graziamaria.virzi@gmail.com; Tel.: +39-0444753650; Fax: +39-0444753949

**Abstract:** Kidney diseases, including acute kidney injury (AKI) and chronic kidney disease (CKD), represent a general, public health urgency, causing a heavy burden to global health care systems. Moreover, AKI is a frequent complication of hospitalization, and it is associated with short-term morbidity and mortality rate greater than 50%, as a result of its relationship with other severe complications. Furthermore, multiple pathophysiologic processes are involved in AKI, such as cellular death, apoptosis, mesenchymal transition, cellular infiltration, inflammation, cytokines release, coagulation, and complement activation. Since increasing evidence highlighted the central role of the immune system in AKI pathophysiology, several efforts have been made to delineate the link between this disease and the different population of immune cells. This narrative review aims to describe the role played by dendritic cells (DCs) in AKI, with a special focus on recent findings suggesting DCs suppression as a promising strategy to prevent AKI negative side effects and ameliorate renal injury and dysfunction. Furthermore, we briefly summarize the main characteristics of innovative high-throughput techniques, including as genomics, transcriptomics, proteomics, and metabolomics in the context of AKI.

**Keywords:** dendritic cells; acute kidney injury; prevention; immune system; renal injury





## 1. Introduction

Acute kidney injury (AKI) and chronic kidney disease (CKD) represent a general, public health urgency, with high impact on global health care systems [1]. AKI, defined as a rapid (within 7 days) increase in serum creatinine, deceased urine output, or both [2,3], is a frequent complication of hospitalization. It is associated with increased short-term morbidity and mortality rate, until 50%. This is the result of pathogenetic consequences of the acute loss of kidney [1], such as fluid overload, acid-base and electrolyte disorders, and progression to CKD [4] and severe associated comorbidities [2]. Moreover, AKI is not a single disease but rather a collection of several syndromes, where the kidney injury is only the consequence of a systemic severe state.

Although many efforts have been made for a worldwide accepted definition and staging [3], for a deeper understanding of its pathophysiology, and for an early diagnosis through novel biomarkers [5], AKI still represents an actual clinical challenge. Ongoing studies are focusing on the identification of specific therapies with impact on morbidity and mortality and on kidney function recovery.

## 2. AKI: Definition and Etiology

Glomerular filtration rate (GFR) is the most useful index of kidney function. In clinical practice, an abrupt decline in GFR is assessed from an increase in serum creatinine (SCr)

or decreased urine output. According to KDIGO guidelines, AKI is defined as an increase in SCr by 0.3 mg/dL within 48 h; or increase in SCr to 1.5 times baseline, which is known or presumed to have occurred within the prior 7 days; or urine volume of 0.5 mL/kg/h for 6 h [3,6]. It includes several conditions affecting kidney structure and function, encompassing various etiologies: specific kidney diseases (e.g., acute interstitial nephritis, acute glomerular and vasculitic renal diseases); non-specific conditions (e.g., ischemia, toxic injury); and extrarenal pathology (e.g., prerenal azotemia, and acute postrenal obstructive nephropathy). More than one of these conditions may coexist in the same patient [6].

According to its pathophysiology, AKI can be classified into three categories: pre-renal, intrinsic, and post-renal [6]. Table 1 summarizes the main mechanisms of AKI. In the hospital setting, pre-renal AKI and acute tubular necrosis (ATN) account for the majority of AKI cases.

**Table 1.** Ethiopatogenesis of AKI.

| Pathophysiology | Mechanism | Causes | Main Examples |
|---|---|---|---|
| Pre renal | Reduced RBF | Absolute hypovolemia | - Blood loss (hemorrhage)<br>- Gastrointestinal loss (vomiting, diarrhea)<br>- Renal loss (diuretic, osmotic diuresis)<br>- Extracellular sequestration (acute pancreatitis, muscle trauma) |
| | | Relative hypovolemia | - Cardiac failure (cardiorenal syndrome)<br>- Peripheral vasodilatation (sepsis, hepatorenal syndrome) |
| | | Impaired blood supply | - Renal artery occlusion (stenosis, thrombosis)<br>- renal vein supply (thrombosis) |
| Intrinsic | Glomerular damage | Glomerulonephritis | - Primary (e.g., FSGS, MCD, membranous nephropathy, IgA nephropathy)<br>- Secondary to systemic disease, autoimmune (lupus nephritis), infectious (post infectious, endocarditis), hematological (MGRS) |
| | Tubulointerstitial damage | Acute tubular necrosis | - Ischemic insults<br>- Exogenous nephrotoxins (drugs, contrast media, toxic)<br>- Endogenous nephrotoxins (myoglobin, hemoglobin) |
| | | Acute interstitial nephritis | - Drugs (NSAIDs, antibiotics)<br>- Infection<br>- Systemic disease (connetivitis) |
| | Intravascular damage | Small-vessel disease | - Trombotic microangiopathy<br>- Renal atheroembolism<br>- small-vessel vasculitis (ANCA-vasculitis, anti-GBM) |
| Post renal | Intratubular obstruction | Protein cast precipitation | - Paraprotein (myeloma cast nephropathy)<br>- Hemoglobin, myoglobin |
| | | Crystals precipitation | - salt overproduction (phosphate, oxalate, uric acid)<br>- drugs (methotrexate, acyclovir, sulfonamides, indinavir, triamterene |
| | Urinary tract obstruction | Kidney stones | - Urethral, bilateral pelvic/ureteral obstruction<br>- ureteral obstruction in single kidney |

**Table 1.** *Cont.*

| Pathophysiology | Mechanism | Causes | Main Examples |
|---|---|---|---|
| | | Prostatic/bladder disease | - benign prostatic hypertrophy<br>- prostatic cancer<br>- urothelial cancer |
| | | Extrinsic compression | - intra-abdominal mass<br>- retroperitoneal fibrosis |

RBF, renal blood flow; FSGS, focal segmental glomerular sclerosis; MCD, minimal change disease; MGRS, monoclonal gammopathy of renal significance; NSAIDs, non-steroidal anti-inflammatory drugs; ANCA, anti-neutrophil cytoplasmic antibody; GBM, glomerular basement membrane.

Pre-renal AKI is mainly secondary to intravascular volume reduction which results in renal blood flow (RBF) impairment. It could be the consequence of net body fluid reduction, such as hemorrhage, gastrointestinal losses (diarrhea, vomiting, prolonged nasogastric drainage), renal losses (diuretics, osmotic diuresis), dermal losses (burns, extensive sweating), or from sequestration of fluid in body tissues, so-called "third spacing" (e.g., acute pancreatitis, muscle trauma). Renal perfusion may be moreover impaired even in the setting of normal or increased extracellular fluid, for example, when cardiac output is decreased (e.g., heart failure) or when cardiac output is distributed to extrarenal vascular beds, as a consequence of systemic arterial vasodilation (e.g., sepsis, liver cirrhosis) or when vascular supply is impaired (renal artery or vein thrombosis). Pre-renal AKI can be reversed with RBF restoration; lack of tempestive intervention will lead to ischemic ATN and tubular cell injury. Intrinsic AKI may be the consequence of renovascular disorders (vasculitis, thrombotic microangiopathies), glomerular, tubular or interstitial diseases. Post-renal AKI could be divided into intratubular and extrarenal forms. To the first group belong tubular precipitation of insoluble crystals (e.g., phosphate, oxalate, uric acid, methotrexate, acyclovir, sulfonamides, indinavir, triamterene) or protein (hemoglobin, myoglobin, paraprotein). The resulting increased intratubular pressure opposes glomerular filtration pressure and can decrease GFR. Extrarenal forms include obstruction of renal pelvis, ureters, bladder, or urethra and are common in patient with prostatic disease, urinary stones, single kidney or intra-abdominal mass.

## 3. Acute Kidney Injury and Immune System

A close cross-talk between immune system and kidney has been largely described. Renal epithelial tubular cells damage and immune system activation, both innate and adaptative, are involved in the pathogenesis of intrinsic AKI. Several pathways could be simultaneously activated, including oxidative stress, complement system, adhesion molecules, and different cell populations could be involved such as macrophages, neutrophils, lymphocytes, and natural killer T resident renal dendritic cells (DCs), contributing to the acute renal damage. Renal DCs are located in the tubulointerstitium and they are involved in T-cell antigen presentation [7]. It has been proven that DCs are the main source of TNF secretion within the first 24 h in ischemic kidney injury [8]. Moreover, they are involved in the recovery after ischemic/reperfusion renal injury, due to their intrinsic property of shift between pro-inflammatory and anti-inflammatory phenotypes [9]. This complex DCs renal network plays a pivotal role in the pathogenesis of intrinsic AKI.

Sepsis-associated AKI development is an example of intensive cross-talk between immune system and kidney. Its pathogenesis is mostly multifactorial and involved an heterogeneous population of patients with different genetic aspects, age, renal function, and comorbidities [10,11]. On the one hand numerous molecular processes could be identified, including cellular death, apoptosis, mesenchymal transition, cellular infiltration, inflammation, cytokines release, coagulation, and complement activation [11], on the other hand AKI itself increases the risk of developing an infection, as a consequence of the

imbalance of cytokines production and clearance, and the impairment of some immune cells function, such as neutrophils [12].

Since increasing evidence highlighted the central role of the immune system in AKI pathophysiology, deeper knowledge about the link between AKI and specific population of immune cells represents an interesting and promising field of research.

### 4. Aims of the Work

This narrative review aims to describe the role played by DCs in AKI, with a deeper focus on recent findings suggesting DCs suppression as promising strategy to prevent AKI negative side effects and to ameliorate renal injury and dysfunction. Furthermore, we briefly summarize the main features of innovative high-throughput techniques, such as genomics, transcriptomics, proteomics, and metabolomics in the context of AKI.

### 5. Material and Methods

A complete research in PubMed and Cochrane databases was carried out by the following search strings: (acute kidney injury OR (acute kidney damage)) AND (dendritic cell OR immune cell response OR immune cell), (acute kidney injury OR (acute kidney damage)) AND (dendritic cell), (immune cell therapy AND AKI therapy) (dendritic cell AND AKI therapy). Furthermore, PubMed was used to identify narrative or systematic reviews and published using specific terms to elaborate and add details to our results. The references of the retrieved papers were used to find more literature. In Figure 1, we reported exclusion criteria for papers and first choice criteria for article selection (Figure 1). Furthermore, we expanded Bibliographic research using narrative review about -omics techniques and biomarkers in AKI.

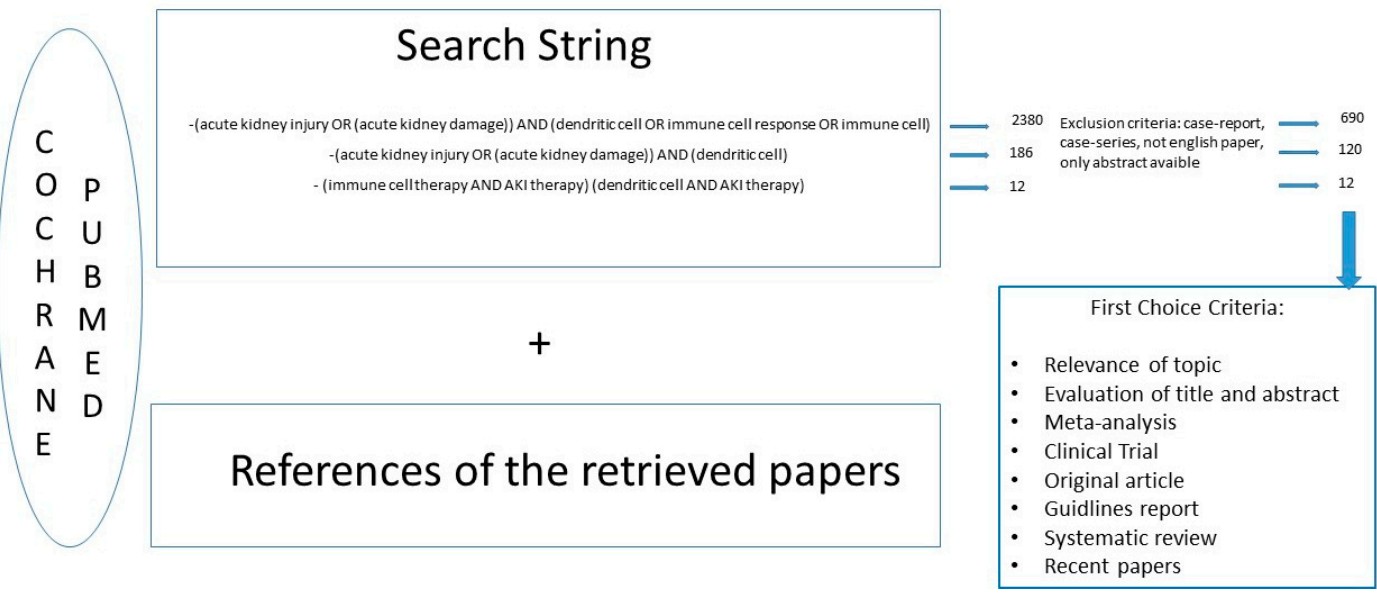

**Figure 1.** Graphical summary of search criteria in the literature.

### 6. Immune Cells in the Kidney

The immune system is fundamental to preserving renal homeostasis by means of numerous resident or circulating effector cells. At least 18 types of cells are present in human kidney, arranged and located in specific structures and areas of the kidney tissue [13]. This organization allows renal specialized physiologic functions and manages the answer

to damage and injury [14]. Different immune cells are located across the renal interstitium, and a deep cellular crosstalk exists between them and the specialized renal cells (e.g., renal tubular epithelial cells) [15–17]. Resident and infiltrated immune cells are crucial in the maintenance of renal homeostasis. Furthermore, immune cells take part in surveillance and defense against infection (immune monitoring), tolerance of non-harmful antigens, regulation of immune-mediated inflammatory pathways, maintaining infection defense capability and repair processes [15,16,18–21]. They include resident phagocytes, such as macrophages and dendritic cells, and infiltrated cells, such as neutrophils and lymphocytes, that might be recruited and enter the kidney from the circulation in response to renal triggers [16,19,22].

## 7. Immune Cells in Acute Kidney Injury

Current studies underlined the role of both innate and adaptive immune responses to endogenous factors released in case of tissue damage or infection [23–25]. During infection or inflammatory conditions, innate immune system is promptly triggered by a non-antigenic-specific way, with a primary pro-inflammatory response promoted by myeloid cells, neutrophils, granulocytes, monocytes, macrophages, DCs, natural killers (NK), and natural killer T cells (NKT) [23,24]. Adaptive immunity, instead, is a secondary line of defense that includes cells mediating an antigen-specific response, such as B lymphocytes and helper- ($T_H$), killer- ($T_C$), and regulatory- (Treg) T cells [26].

In this context, immune effector cells, such as DCs and macrophages, serve both as mediator cells between innate and adaptive immunity, producing cytokines, and chemokines, but also directly as presenting-antigens cells to lymphocytes [27] (Figure 2). Notably, signaling cascades activated by intrinsic and extrinsic factors merge to a singular pathway, as activation of both innate and adaptive immune responses is regulated by the Toll-like receptor (TLR) pathways [25,28].

**Figure 2.** Graphical summary of immune system response.

Different immune cell populations have been described to contribute to AKI pathophysiology [12,29]. In fact, the pro-inflammatory role of DCs, monocytes/M1 macrophages, neutrophils, T and B lymphocytes in exacerbating renal injury and dysfunction is well-known. However, some immune cells, including M2 macrophages, DCs, and regulatory T cells (Treg), could also show an opposite effect on AKI, suppressing inflammation and promoting tissue remodeling and repair. Given their bi-directional role, a role of DCs as modulator to prevent and ameliorate AKI negative side effects is emerging.

## 8. Dendritic Cells and the Kidney

DCs are formidable antigen-presenting cells, originating in the bone marrow from both myeloid and lymphoid precursors [30], closely interacting with other immune cells, including CD4$^+$ and CD8$^+$ T lymphocytes, Th, Tregs, and NK cells. DCs act as sentinel system, with a dual role in controlling the immune system, activating an early immune response, and the blockage of co-stimulatory signals, with consequent induction of immunological tolerance [31,32]. Activation and maturation of DCs are linked to "danger signals", including damage-activated molecular patterns (DAMPs), pathogen-activated molecular patterns (PAMPs), TLR signaling, production of cytokines (such as TNF-$\alpha$, IL-6, IL-1$\beta$, IL-23, IL-17A) and IFN-$\gamma$, reactive oxygen species (ROS), and co-stimulatory proteins (i.e., CD80, CD86, etc.) (Table 2). Since the immune response can also be modulated by soluble factors, including complement system, cytokines, chemokines, and TLRs 9 14 15, DCs represent a fundamental link between the two types of immunity [33].

**Table 2.** Summary of DCs activity.

| DENDRITIC CELLS |
|---|
| - Antigen-presenting cells |
| - Link between the innate and adaptive immunity |
| - Interact with CD4$^+$ and CD8$^+$ T lymphocytes, Th, Tregs, and NK cells |
| - Initiation of an early immune response |
| - Blockage of co-stimulatory signals with consequent induction of immunological tolerance |
| - Response to danger signals |

DCs are the most abundant innate immune population in the kidney, where they constantly probe the environment [34]. They take part in kidney homeostasis, and are activated during inflammatory conditions (Figure 3). When an antigen is recognized, DCs trigger an inflammatory response through the direct release of cytokine and co-stimulatory molecules that recruit neutrophilic granulocytes into the renal district [35] and the antigen-presentation to naïve T cells [36]. Notably, as over-activation of inflammatory responses can induce tissue damage, DCs also contribute to the maintenance of organ integrity, generating tolerance through induction of autoreactive T cell or expanding Treg apoptosis [37]. Moreover, also in healthy kidney, DCs induce apoptosis of T cells specific to self-antigen or innocuous small molecular weight antigens, to guarantee immunologic tolerance [36,38].

## DENDRITIC KIDNEY CELLS

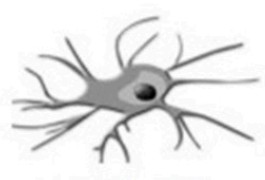

| HOMEOSTASIS: | PRO-INFLAMMATORY: | ANTI-INFLAMMATORY: |
|---|---|---|
| - Induction of autoreactive T cells apoptosis; | - Secretion of inflammatory citokines; | - Suppression of inflammatory citokines; |
| - Immune tolerance against renal autoantigens; | - Uptake and presentation of non-filtrable antigen to T cells; | - Limiting renal tissue damaging; |
| - Immune tollerance against filtrable antigens. | - Promotion of fibrosis; | - Limiting nephrotoxic nephritis. |
| | - Recruitment of neutrophilic granulocytes. | |

**Figure 3.** Renal DCs properties and main activity.

The number of DCs increases during inflammation following renal ischemia-reperfusion injury (IRI) [39]. Also in IRI condition, DCs have been shown to have a dual role, promoting inflammation through the production of pro-inflammatory chemokines and cytokines and preventing excessive tissue damage via anti-inflammatory mechanisms [9,40]. In AKI, renal endothelial cells (REDs) have been described to be a central target of DCs, such that DCs-REDs interaction is one of the first steps in the IRI. In fact, REDs injured by IR are particularly vulnerable to neutrophils and DCs adhesion, migration, and activation, with consequent release of cytokines, chemokines, and growth factors that promote inflammation. Particularly, TNF-α released by DCs during the first 24 h after IRI, binds REDs, inducing apoptosis and promoting leukocyte extravasation [41]. Of note, TNF-α levels have been demonstrated to correlate with the grade of kidney damage [42,43]. In this context, the role of DCs as modulator to silence unwanted immune responses has been proposed [26]. Recently, Chopra et al. have reviewed different pharmacological agents and measures to prevent AKI, providing novel targets and cell-based therapy (CD4$^+$CD25$^+$ T cells, Treg, and DCs) for this purpose [44]. Although a real AKI- preventive strategy based on DCs targeting is still lacking, the concept of inhibition of DCs maturation and function to abrogate T-cell activation and proliferation has emerged [44–48] and has been tested in some animal studies [49,50]. The modulation of DCs to prevent renal injury has been postulated through different approaches, such as inhibition, blockage, suppression, tolerance, depletion, deletion, abrogation.

## 9. Inhibition of Dendritic Cells Function to Prevent Renal Injury

In 1999, Verhasselt et al. investigated for the first time the influence of N-acetyl-l-cysteine (NAC), an antioxidant molecule, on human DCs, evaluating NF-κB activity, cytokine production, and expression of co-stimulators both at the basal state and after activation by either LPS or CD40 engagement. The authors reported several NAC-mediated consequences on DCs: (i) Inhibition of both constitutive and LPS-induced NF-κB activation; (ii) down-regulation of cytokines production and DCs-specific surface markers expression (i.e., HLA-DR, CD86, and CD40) both at the basal condition and after activation with LPS; (iii) inhibition of DCs responses following CD40 engagement. All these findings revealed NAC was able to indirectly inhibit the primary T-cell responses through its immunosuppressive action on DCs, offering a potential strategy to prevent undesirable immune responses [49]. In 2006, NAC capability to inhibit DCs was further studied by Di Giorno et al. in mice receiving renal IR, mimicking human AKI. Animals have been divided into two groups, each one receiving a different concentration of NAC before surgery, and renal function, acute tubular necrosis and immune-phenotyping of infiltrating cells have been evaluated. Interestingly, mice treated with higher dose of NAC had lower serum creatinine levels and lower acute tubular necrosis with reduced renal DCs infiltration, showing NAC pretreatment was beneficial in limiting IR-induced damage [50]. A similar effect in preventing IRI was described by Okusa et al., when testing a selective agonist of adenosine 2A (A$_{2A}$) [46,51]. A$_{2A}$ is an important anti-inflammatory molecule released during tissue inflammation, whose concentration can increase more than six-fold after only two minutes of ischemia. The innate immune cells have several types of adenosine-surface receptors (A$_{2A}$-Ars) able to modulate the cellular responses inhibiting inflammation [52]. Okusa's group investigated whether selective activation of A$_{2A}$-Ars was able to reduce renal IRI, testing the A$_{2A}$ agonist DWH-146e in rodent models of ischemic AKI. These models were chosen because of their well-established processes of reperfusion on the inflammatory pathways [46,51,53]. Results showed that DWH-146e was able to protect the kidney from IRI, inducing a reduction of plasma creatinine, even when the compound was administered during reperfusion period [49]. As DWH-146e was described to preserve both renal function and histology [46], additional studies deepened insight into the possible mechanisms involved in A2AR-agonist-mediated protection. Another study showed that DWH-146e administration reduced renal neutrophil accumulation, oxidative activity, and adhesion in mice subjected to IR 34. Additional works focused on the direct effect that

another $A_{2A}$-ARs agonists (ATL313) had in modulating DCs signaling and activation. Notably, mice with $A_{2A}$-Ars-deficient DCs did not respond to ATL313 agonists and had worse kidney injury. On the contrary, administration of DCs treated ex vivo by ATL313 ameliorated kidney IRI of wild-type mice owing to the direct suppression of NKT cell activation. Of note, the fact that DCs could induce both immunity and tolerance according to the microenvironment condition was well established [52], however, the fact that $A_{2A}$-ARs agonist-mediated tolerance was the result of direct suppression of other immune cells function by DCs was a novel finding. Further progress in understanding the side effects associated with DCs suppression has been made by Saiga et al., using a graft-versus-host disease model. Specifically, the authors showed that treatment of mice with a novel DCs-suppressant (NK026680) allows a reduction in mortality, suppresses cytokine production (i.e., TNF-$\alpha$, IFN-$\gamma$, IL-5, -4, and -2), prevents glomerulonephritis and glomerular damage, and decreases the rates of proteinuria, hematuria, and blood urea nitrogen [26]. The mechanisms explaining NK026680-mediated DCs inhibition have been associated with a decrease in major histocompatibility complex (MHC) class I and II, CD83, and CD86 on human monocyte-derived DCs during immune system stimulation and the inhibition of lymphoproliferative T-cell development.

A recent study has revealed both in vitro and in vivo, how overexpression of miR-21 is a self-protective response of the kidney under IR-conditions. Interestingly, this protective role has been explained through the ability of miR-21 to inhibit the maturation of DCs activating PDCD4/NF-$\kappa$B pathway, with consequent reduction of pro-inflammatory molecules released by DCs 38. Once again, DCs activity has been proven to be crucial in determining the severity of kidney IR-induced damages. Similarly to NAC, Paeonol is a molecule with an anti-oxidant activity. In a murine model of LPS-induced kidney injury, a reduction in blood urea nitrogen and serum creatinine has been observed and it attenuates the effects of LPS on dendritic cells, with significant inhibition of pro-inflammatory cytokines release, and suppression of TLR4 expression and NF-$\kappa$B signal pathway [54]. Galectin-3 is a surface molecule of DCs crucial in their activation. In animal model it has been observed that its activation protects from cisplatin-induced AKI by promoting TLR-2-dependent activation of IDO1/KYN pathway in renal DCs, resulting in an increased expansion of immunosuppressive Tregs in injured kidneys [55]. Administration of FTY720, a synthetic sphingosine-1-phosphate, attenuated IRI kidney and reduced inflammation. The beneficial effect of FTY720 was associated with the expansion of peripheral CD11b(+) CD11c(+) DC and with maturation of spleen CD11c(+) DC, which showed impaired allostimulatory capacity. Beneficial effects of FTY720 in IRI may be partially mediated by DC modulation or by increasing Treg activity [56].

Overall, all these studies support the idea that manipulation of DCs function may be a promising approach to limit the negative side effects associated with immunological disorders, secondary to ischemic insults or toxic insults [11,33,52], and this strategy could be applied to kidney injury, both in native and transplanted kidney. Although these observations are limited to animal models and their application in humans is not so straightforward, directly targeting DCs with therapeutic interventions in AKI seems to be promising.

## 10. A Possible Role of Matrix Metalloproteinases (MMPs) and Tissue Inhibitors of Metalloproteinases (TIMPs) in the Activation of Dendritic Cells

Matrix metalloproteinases (MMPs) are enzymes able to degrade [57] and remodel extracellular matrix components [58], thus driving cell migration [59,60]. Their activity is regulated by specific tissue inhibitors, called TIMPs (tissue inhibitors of metalloproteinases) [61].

Monocyte-derived dendritic cells produce active MMPs and TIMPs, and in particular, elevated migratory capability is connected to MMP-9, which can be inhibited by TIMPs [62]. Therefore, MMP/TIMP imbalance plays a central role in the control of inflammatory mecha-

nisms and cell migration, which characterize pathological conditions such as osteoarthritis, atherosclerosis, and cancer.

Thus, we can hypothesize a possible effect of MMP/TIMP imbalance in the activation of DC in the setting of acute kidney injury as well. TIMP-2 is a recognized marker of kidney injury, together with IGFBP7 (insulin growth factor binding protein 7). We can speculate that increased level of TIMP-2 may represent a response of the body to the MMPs activation, in order to reduce DC migration, thus limiting renal injury. The development of synthetic MMP inhibitors may reduce matrix remodeling, thus modulating inflammatory mechanisms, also implicated in the pathogenesis of acute kidney injury. Further studies are necessary to better comprehension of the role of MMP/TIMP in the regulation of DCs migration, particularly in the setting of inflammatory conditions, such as acute kidney injury.

## 11. Omics Technology for AKI Diagnosis and Prevention

The prevention of AKI is still a challenge, and therapeutic strategies for AKI are mostly supportive and nonspecific. In this context, AKI imperatively needs more impactful and useful prevention and therapeutic interventions.

Recently, research in omics technologies and omics methodologies, ranging from genomics to metabolomics, has been improved. These innovative approaches investigate biological molecules, networks, pathways, and final products on a huge scale, using new generation instruments and computational and bioinformatics approaches.

A comprehensive strategy and integration of multiomic approach could offer a deeper understanding of molecular dynamics and pathways. This strategy could offer the possibility of new and individualized approaches for early detection, specific treatment, and effective prevention on an individual basis.

Given the multifaceted features of AKI, -omic analysis, together with clinical data and disease features, can be helpful to recognize and identify molecules involved in the mechanism that could be used as biomarkers (for example: prognostic biomarkers, biomarkers associated with therapy, prediction biomarkers, and biomarkers to develop precision medicine) [59].

In order to ameliorate patient's outcome, recognition of biomarkers to identify AKI progression, to evaluate the need of renal replacement therapy, and to predict the response to therapy, as well as the grade of renal regeneration or residual CKD post-AKI, would be useful. Recently, many novel biomarkers have been described as early detectors of AKI. Unfortunately, the majority of those are associated only and strictly to a single physiopathologic mechanism, for example tubular injury. This observation can clarify why these markers often have failed in AKI secondary to other pathological processes or with heterogeneous origin [63].

Recently, the introduction of proteomics and metabolomics highlighted how a single isolated molecule cannot entirely explain the multifaced pathogenesis of AKI. In this context, integrating different markers in one biomarker panel could be a good and desirable goal [58]. Numerous biochemical methodologies are used for proteomics research.

The identification of novel biomarkers for a promptly and early diagnosis of AKI is a promising field of coming research. An ideal biomarker for AKI should meet different requirements. For example, it should be appropriate in pre-analytical setting, it should be specific and sensitive, accurate, stable, reliable, standardized, fast, and cheap.

Unique and comprehensive biomarker for AKI will not capture all clinical conditions caused by AKI [63]. Promising results about -omics application for AKI are mainly available in the context of sepsis-associated AKI (SA-AKI), with biomarkers aiming to predict the risk of AKI development [64–66] or worse outcomes [67–69]. Other examples include biomarkers in the context of AKI after cardiac surgery [70,71], and urinary markers [72–74] or plasmatic markers associated with renal recovery [75,76]. A complete panel with multiple and appropriate AKI biomarkers could be successfully applied and improve performance for AKI diagnosis and could be more sensitive, specific, and predictive with respect to the

single marker evaluation system [77,78] and could be useful in early identification of AKI complications, including involvement of other organs (cardio-renal syndrome, hepato-renal syndrome, etc.).

## 12. Precision Medicine for Acute Kidney Injury (AKI)

The current progresses in omics technologies permit a more comprehensive understanding of AKI pathophysiology and processes. Improvement in this field could allow the introduction of precision medicine in the clinical approach to AKI. *Precision medicine* is an emerging subject with the aim to match treatment and prevention as closely as possible to single patient features, based on individual biological, genetic, environment, lifestyle, and clinical variability. Tailored AKI management would be achieved if multiple phenotypes and sub-phenotypes of AKI would be identified. In particular, molecules associated with specific AKI sub-phenotypes should be detected and specific biomarkers could highlight specific pathways and networks. Besides an early diagnosis, stratifying AKI population in subgroups with specific clinical, biological, and molecular patterns would help clinician to predict early response to therapy [79,80] and the cases associated with worse prognosis [71,81]. Chances for employing this method have increased significantly in the past years, based on the huge amount of information for each subject in genetic (Genomics, Epigenomics and Transcriptomics) and in molecular setting (Proteomics and Metabolomics) [82–84].

## 13. Conclusions

In conclusion, this narrative review describes the role of DCs during AKI and their putative clinical implications, suggesting DCs suppression as a promising strategy to prevent AKI and its complications, and ameliorating renal injury and dysfunction. Nowadays evidence on human subject are still lacking, and further data will be necessary for a deeper knowledge of the role of DCs in AKI, for optimized biomarkers for their identification, and for future employment of DCs in routine clinical practice.

Furthermore, we briefly summarized new aspects of -omic techniques and precision medicine in the setting of AKI. Emerging details and continuous research on novel advanced methods could suggest original diagnostic options for AKI diagnosis, evaluation, and prevention, owing to the identification of novel biomarkers. With the help of these promising technologies, an optimized biomarker assessment and multiple biomarker panel for AKI supporting the clinical decision, influencing time to therapeutic intervention and improving patient's outcome could be achieved.

**Author Contributions:** G.M.V. and A.C.: conception of the study, drafting of the article, revision of the article; M.M.: rafting of the article and revision of the article; G.G.B. and C.R.: final approval of the article and M.Z.: providing intellectual content of critical importance to this work. All authors have read and agreed to the published version of the manuscript.

**Funding:** This research received no specific grant from any funding agency in the public, commercial, or not-for-profit sectors.

**Institutional Review Board Statement:** Not applicable.

**Informed Consent Statement:** Not applicable.

**Data Availability Statement:** Not applicable.

**Conflicts of Interest:** The authors have no conflict of interest to declare. Prof. Claudio Ronco in the last three years has been consultant, medical advisor, or part of the speaker bureau receiving fees from the following companies: Asahi Medical, Aferetica, Baxter, B. Braun, Biomerieux, Bioporto, Cytosorbents, ESTOR, Fresenius Medical Care, GE Healthcare, Kaneka, Medica, Medtronic-Bellco, Nipro, Spectral, Toray, Jafron.

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
