# Peer review of "New Frontiers in Diagnosis and Prevention of Acute Kidney Injury (AKI): The Role of Dendritic Cells and Innovative High-Throughput Techniques"

_applsci, doi:10.3390/app13074276_

Round 1

Reviewer 1 Report

In the manuscript, entitled »New frontiers in diagnosis and prevention of acute kidney injury (AKI): the role of Dendritic cells and innovative high-throughput techniques« submitted to journal of Applied Sciences for potential publication, the authors present their research article investigating the role of  Dendritic cells in acute kidney injury. Although the topic is of considerable interest, I am of opinion, that in the present form, it is not good enough to be published. My suggestions and comments to improve it are presented below.

The comments:

1.       In general, the article is quite long and difficult to read due to complexity of the investigated field as well as lack of the presentation clearness.

2.       Basic information about acute kidney injury should be stated in the Introduction, including definition and etiology.

3.       The type od review should be defined and argued.

4.       Figure, presenting the pathophysiology of acute kidney injury, would be of value. The same should be done for the role of Dendritic cells.

5.       Table of review articles presenting the research area should be shown.

6.       The studies investigating application of omics technologies in acute kidney injury should be presented.

7.       The multipanel biomarkers of acute kidney injury should be commented.

8.       In addition to precision medicine definition its implication in acute kidney injury should be presented.

9.       Treatment options based on the pathophysiology have to be mentioned as well as some future perspectives.

Author Response

Comments and Suggestions for Authors

In the manuscript, entitled »New frontiers in diagnosis and prevention of acute kidney injury (AKI): the role of Dendritic cells and innovative high-throughput techniques« submitted to journal of Applied Sciences for potential publication, the authors present their research article investigating the role of Dendritic cells in acute kidney injury. Although the topic is of considerable interest, I am of opinion, that in the present form, it is not good enough to be published. My suggestions and comments to improve it are presented below.

The comments:

  1. In general, the article is quite long and difficult to read due to complexity of the investigated field as well as lack of the presentation clearness.

We apologise if the text seemed too long and sometimes not so clear. We performed some modification with the aim to make each section clearer. We deleted some paragraphs. Unfortunately, the minimum number of words depends on the type of manuscript submitted.

  1. Basic information about acute kidney injury should be stated in the Introduction, including definition and etiology.

We complete the initial paragraph with more detailed information regarding AKI, including definition and etiology (line 67-109). In order to make more feasible the main mechanisms at the basis of AKI, we add a summary table (Table 1).

  1. The type of review should be defined and argued.

We thank the Reviewer for the prompt suggestion. Our work is essentially a narrative review, and we explicate it in the abstract and in the method section of the paper (line 147).

  1. Figure, presenting the pathophysiology of acute kidney injury, would be of value. The same should be done for the role of Dendritic cells.

The pathophysiology of AKI has been summarized in table 1 because we found it more immediate for a complete synthesis. Dendritic cells role in kidney tissue has been showed in Figure 3.

  1. Table of review articles presenting the research area should be shown.

In figure 1, we summarize the criteria of inclusion and exclusion of our review in this work and our results. The references of the included papers have been reported at the end of the manuscript, in the bibliography section (line 484-784). We did not draft any table of review in order to give more space to possible application and clinical implications of these knowledges and because we found it somehow redundant. We hope our choice could be understood by Reviewer. If it would not, we are available to complete our work with this content.

  1. The studies investigating application of omics technologies in acute kidney injury should be presented

We add some references (line 421-427) of original articles as examples of omics applications in the filed of AKI and its clinical counterpart. A complete and systematic review of those goes beyond the scope of our work. Therefore, our selection has been limited to the most recent or clinical relevant works, according to our knowledge.

  1. The multipanel biomarkers of acute kidney injury should be commented.

We thank the review for the suggestion. We complete (line 430-432) with comment about putative multipanel biomarkers application and clinical application.

  1. In addition to precision medicine definition its implication in acute kidney injury should be presented.

We add some references about precision medicine implication in AKI (line 445-447). We hope that this section could be enough complete.

  1. Treatment options based on the pathophysiology have to be mentioned as well as some future perspectives.

Ongoing therapeutical approach to AKI is based on etiological treatment, besides supportive measures. Increasing knowledges on molecular pathways activated thanks to omics technologies would offer a wider spectrum of therapeutical options, included in the field of precision medicine, as we stated in our conclusion (line 453-468).

Reviewer 2 Report

Would you be able to present us with the clinical implications and what are the conditions in which the activation of DCs should be suspected?

Author Response

Comments and Suggestions for Authors

Would you be able to present us with the clinical implications and what are the conditions in which the activation of DCs should be suspected?

We thank the reviewer for the comment. We stated in line 348-350 that an activation of DCs has been described in animal model of ischemic-reperfusion injury model, which simulated the typical damage of native and transplanted kidney. Therefore, the main application would be a modulation of their activity in order to limit the tissue injury amplification. Omics technology could be useful to discover specific molecules involved in DCs activity modulation, as promising pharmacological target. We try to explain this concept in the specific paragraph and in our conclusion. We hope the message could be clearer.

Reviewer 3 Report

This is an exciting review focusing on the role of dendritic cells in AKI development and possibilities for ameliorating kidney injury by the influence on DCs. However, in my opinion, this work required substantial corrections:

Major comments:

-          In the Material and Methods section, the Authors specified keywords and search strategies. However, the result of this search was not reported. The authors should refer to how many publications they found useful and how many They reject. This help to introduce the problem's importance to the reader.

-          The section concerning on possible modulating effect of known drugs (other than NAC) on DCs activity is lacking, however, it would be very welcome

-          Although the first part of the manuscript, concerning DCs, is interesting and helpful, the second, describing “-omics” does not bring any solid information and wastes the reader’s time. Please consider short it to one small max. 50-lines part or even remove it completely.

Author Response

Comments and Suggestions for Authors

This is an exciting review focusing on the role of dendritic cells in AKI development and possibilities for ameliorating kidney injury by the influence on DCs. However, in my opinion, this work required substantial corrections:

Major comments:

-          In the Material and Methods section, the Authors specified keywords and search strategies. However, the result of this search was not reported. The authors should refer to how many publications they found useful and how many They reject. This help to introduce the problem's importance to the reader.

We thank the reviewer for the suggestion, we implemented our “material and methods” section and the related image (Figure 1), with details about inclusion and exclusion criteria and number of works analyzed, rejected and excluded from our analysis.

The section concerning on possible modulating effect of known drugs (other than NAC) on DCs activity is lacking, however, it would be very welcome

      Unfortunately a little number of work concerning activity of drug on DCs activity are nowadays available, and they are mostly limited to pre clinical study. We spread our research in this field, and we found some other examples of promising molecules in modulating DCs activity (line 330-345).

-          Although the first part of the manuscript, concerning DCs, is interesting and helpful, the second, describing “-omics” does not bring any solid information and wastes the reader’s time. Please consider short it to one small max. 50-lines part or even remove it completely.

      We summarized the section about -omics (379-432), focusing on the notions related to clinical application in the context of AKI and DCs targeting. We hope our work could be more affordable with these corrections.
